# Flavonoids from *Chionanthus retusus* (Oleaceae) Flowers and Their Protective Effects against Glutamate-Induced Cell Toxicity in HT22 Cells

**DOI:** 10.3390/ijms20143517

**Published:** 2019-07-18

**Authors:** Yeong-Geun Lee, Hwan Lee, Jae-Woo Jung, Kyeong-Hwa Seo, Dae Young Lee, Hyoung-Geun Kim, Jung-Hwan Ko, Dong-Sung Lee, Nam-In Baek

**Affiliations:** 1Graduate School of Biotechnology and Department of Oriental Medicinal Biotechnology, Kyung Hee University, Yongin 17104, Korea; 2College of Pharmacy, Chosun University, Gwangju 61452, Korea; 3Strategic Planning Division, National Institute of Biological Resources, Incheon 22689, Korea; 4Department of Herbal Crop Research, National Institute of Horticultural and Herbal Science, RDA, Eumseong 27709, Korea

**Keywords:** *Chionanthus retusus*, flavonoid, flower, HO-1, neuroprotection, NO

## Abstract

The dried flowers of *Chionanthus retusus* were extracted with 80% MeOH, and the concentrate was divided into EtOAc, *n*-BuOH, and H_2_O fractions. Repeated SiO_2_, octadecyl SiO_2_ (ODS), and Sephadex LH-20 column chromatography of the EtOAc fraction led to the isolation of four flavonols (**1**–**4**), three flavones (**5**–**7**), four flavanonols (**8**–**11**), and one flavanone (**12**), which were identified based on extensive analysis of various spectroscopic data. Flavonoids **4**–**6** and **8**–**11** were isolated from the flowers of *C. retusus* for the first time in this study. Flavonoids **1**, **2**, **5**, **6**, **8**, and **10**–**12** significantly inhibited NO production in RAW 264.7 cells stimulated by lipopolysaccharide (LPS) and glutamate-induced cell toxicity and effectively increased HO-1 protein expression in mouse hippocampal HT22 cells. Flavonoids with significant neuroprotective activity were also found to recover oxidative-stress-induced cell damage by increasing HO-1 protein expression. This article demonstrates that flavonoids from *C. retusus* flowers have significant potential as therapeutic materials in inflammation and neurodisease.

## 1. Introduction

With the rapid growth of the aging population, the treatment of age-related diseases has become an important global issue, including in Korea [1]. Neurodisease is among the various illnesses induced by aging [2]. Previous studies have revealed the neuroprotective activities of bioactive compounds such as alkaloids, sterols, and flavonoids [3,4]. Flavonoids perform various neuroprotective actions, such as suppressing neuroinflammation; protecting neurons; and promoting memory, cognitive function, and learning [5,6]. Given the many experiments demonstrating their neuroprotective effects, these compounds may have therapeutic potential in neurodisease [3,6,7,8,9].

Flavonoids have a phenylchromane (C6-C3-C6) structure and are synthesized from l-phenylalanine and l-tyrosine via the shikimic acid pathway [10]. They comprise one of the most widespread and diverse groups of compounds in nature [11,12,13]. Among various natural resources, flowers (the reproductive organs of plants) contain diverse secondary metabolites, including volatiles, pigments, and flavonoids, which lure pollinating insects and facilitate pollination [14,15,16]. Sun et al. previously determined the total flavonoid content of *Chionanthus retusus* flowers to be 10.7% [17]. Thus, in this study, we focused on the isolation, identification, and investigation of the potential therapeutic effects of flavonoids from *C. retusus* flowers.

*C. retusus* (Oleaceae), a deciduous tree with oval leaves, is widely cultivated and distributed in Korea, China, Taiwan, and Japan, growing to 20–25 m high [18]. This plant has been used as an antipyretic, treatment for palsy and diarrhea in Oriental medicine and is known to contain many kinds of secondary metabolites, including flavonoids, lignans, sterols, and terpenoids [18,19,20]. These compounds have been reported to exert antioxidant, anti-inflammatory, and neuroprotective effects [6,7,18]. Although numerous active components have been isolated from *C. retusus* leaves and stems, the flowers of *C. retusus* have rarely been studied. This paper describes the isolation of 12 flavonoids from *C. retusus* flowers, determination of their chemical structures through extensive analysis of various spectroscopic data, evaluation of their anti-inflammatory and neuroprotective effects, and the relationship of their structure to their activity.

## 2. Results and Discussion

### 2.1. Contents of Total Phenols and Total Flavonoids in C. retusus Flowers

The contents of total phenols and flavonoids in the extract and fractions were determined as gallic acid and catechin equivalent values, respectively. As shown in Table 1, MeOH extract and EtOAc fraction (fr.) showed the highest contents compared to other fr.s. MeOH extract and EtOAc fr. showed a yellowish color on a thin-layer-chromatography (TLC) plate by spraying 10% H_2_SO_4_ and baking (data not shown), suggesting the extract and EtOAc fr. to include high amounts of flavonoids.

### 2.2. Isolation and Identification of Flavonoids from C. retusus Flowers

The dried flowers of *C. retusus* were extracted with MeOH, and the concentrate was divided into EtOAc, *n*-BuOH, and H_2_O fr.s. Repeated SiO_2_, octadecyl SiO_2_ (ODS), and Sephadex LH-20 column chromatography (c.c.). on the EtOAc Fr enabled the isolation of four flavonols (**1**–**4**), three flavones (**5**–**7**), four flavanonols (**8**–**11**), and one flavanone (**12**). These compounds were identified to be quercetin (**1**) [20], kaempferol (**2**) [20], astragalin (**3**) [21], nicotiflorin (**4**) [22], luteolin (**5**) [20], luteolin 4′-*O*-*β*-d-glucopyranoside (**6**) [23], isorhoifolin (**7**) [24], taxifolin (**8**) [25], aromadendrin (**9**) [20], aromadendrin 7-*O*-*β*-d-glucopyranoside (**10**) [26], taxifolin 7-*O*-*β*-d-glucopyranoside (**11**) [27], and eriodictyol 7-*O*-*β*-d-glucopyranoside (**12**) [24] based on extensive analysis of data from various spectroscopic methods, including IR, FAB/MS, 1d-NMR (^1^H, ^13^C, DEPT), and 2d-NMR (COSY, HSQC, HMBC) (Figure 1). The identities of the compounds were confirmed by comparing their NMR and MS values with those reported in the literature. We determined the stereochemistry of the chiral centers (C-2 and C-3) in flavonoids **8**–**12** by examining the coupling constants between H-2 and H-3 in the ^1^H-NMR spectra. They were mostly observed to be 12 Hz, which suggested that the two protons were in a 2,3-*trans* configuration.

### 2.3. Inhibition Effects of Flavonoids ***1***–***12*** on NO Production in Lipopolysaccharide (LPS)-Induced RAW 264.7 Cells

Oxidative stress is not only an important feature of several neurodegenerative processes, but also actively triggers intracellular signaling pathways that lead to cell death [28]. We first examined the viability of RAW264.7 cells treated with compounds **1**–**12** using a 3-(4,5-dimethylthiazol-2-yl)-2,5-diphenyltetrazolium bromide (MTT) assay. It did not show cytotoxicity or cellular proliferation when treated with compounds **1**–**4** and **6**–**12** at concentrations of 40 or 80 µM in RAW264.7 cells. However, compound **5** exhibited cytotoxic effects at 80 µM (Figure 2a). To investigate the anti-inflammatory effects of compounds **1**–**12**, we appreciated their inhibitory effects on NO production in LPS-induced RAW 264.7 cells. These cells were pretreated with flavonoids **1**–**12** and butein, a positive control, before one day LPS treatment. As shown in Table 2, compounds **1**, **2**, **5**, and **6** highly inhibited NO production, while compounds **8** and **10**–**12** showed moderate inhibition effect. Flavonoids with a catechol structure in the B ring (**1**, **5**, **6**, **8**, **11**, and **12**) exerted stronger anti-inflammatory effects than those with a phenol structure (**3**, **4**, **7**, **9**, and **10**). In addition, as the number of glucose moieties increased in compounds **1**–**6**, the NO inhibitory effects of these compounds in RAW 264.7 cells decreased. However, compounds with glucopyranosyl moieties at C-7 (**10** and **11**) exhibited higher activity than aglycones (**8** and **9**). These results indicate that the presence of a catechol structure in the B ring and a glucopyranosyl moiety in the flavonoid structure were key factors of the anti-inflammatory effects of these flavonoids.

### 2.4. Effects of Flavonoids ***1***–***12*** on Glutamate-Induced Cell Toxicity in Mouse Hippocampal HT22 Cells

To investigate the protective effects of compounds **1**–**12** against glutamate-induced oxidative neuronal cell death, we also examined their effects on the viability of mouse hippocampal HT22 cells. To investigate the potential for cellular proliferation or cytotoxic effects of compounds **1**–**12**, we first examined the viability of mouse hippocampal HT22 cells treated with compounds **1**–**12** using an MTT assay. No cytotoxic effects or cellular proliferation by compounds **1**–**12** were observed at concentrations <40 µM (Figure 2b). These cells were pretreated with compounds **1**–**12** at concentrations of 20 or 40 μM for 3 h and then were treated with glutamate and reacted for 12 h. Thereafter, cell viability was assessed with an MTT assay. None of the compounds exhibited toxicity at the highest concentration (40 μM). Compounds **1**, **2**, **5**, **6**, **8**, **10**, **11**, and **12** significantly increased cell viability following glutamate treatment (Figure 3). Butein derived from *Rhus verniciflua*, which is known to protect mouse hippocampal HT22 cells from glutamate-induced death [29], was used as a positive control and indeed exhibited cytoprotective effects (Figure 3). Flavonoids with a catechol structure in the B ring (**1**, **5**, **6**, **8**, **11**, and **12**) exerted stronger cytoprotective effects than those with a phenol structure (**3**, **4**, **7**, **9**, and **10**). In addition, as the number of glucose moieties increased in compounds **1**–**6**, the cytoprotective effects of these compounds in HT22 cells decreased. However, compounds with glucopyranosyl moieties at C-7 (**10** and **11**) exhibited higher activity than aglycones (**8** and **9**). These results indicate that the presence of a catechol structure in the B ring and a glucopyranosyl moiety in the flavonoid structure were key determinants of the effects of these flavonoids on mouse hippocampal HT22 cells.

### 2.5. Effects of Compounds ***1***, ***2***, ***5***, ***6***, ***8***, and ***10***–***12*** on HO-1 Expression in Mouse Hippocampal HT22 Cells

Heme oxygenase (HO) is an important enzyme in the antioxidant cell system. HO-1, one of the HO derivatives, decomposes heme in the cell to produce carbon monoxide, iron, and biliverdin [30]. HO-1 expression has been reported to inhibit brain cell damage resulting from oxidative stress [31]. We examined whether compounds **1**, **2**, **5**, **6**, **8**, and **10**–**12** affected the protein expression of HO-1, given their protection against glutamate-induced toxicity in mouse hippocampal HT22 cells. Mouse hippocampal HT22 cells were treated with compounds **1**, **2**, **5**, **6**, **8**, and **10**–**12** at three concentrations (10, 20, and 40 μM) and then cultured for 12 h. Cobalt protoporphyrin (CoPP), a well-known HO-1 inducer, was used as a positive control. As shown in Figure 4, compounds **1**, **2**, **5**, **6**, **8**, and **10**–**12** all increased HO-1 protein expression in a dose-dependent manner in mouse hippocampal HT22 cells. Flavonoid aglycones (**1**, **2**, **5**, and **8**) exhibited higher activity than the glycosides (**10**–**12**). The flavonol and flavanonol with a catechol structure in the B ring (**1** and **11**) displayed stronger HO-1 expression than those with a phenol structure (**2** and **10**). Flavonoids with a hydroxy group at C-3 (**8** and **11**) exhibited weaker HO-1 expression than those without (**5** and **12**). In addition, a flavonoid with a double bond between C2 and C3 (**1**) was a weaker inhibitor of oxidative-stress-induced brain-cell damage than one with a single bond (**8**). These results indicate that the presence of a hydroxy group at C-3, the structure of the B ring and the type of C2‒C3 bond are key determinants of the extent to which these flavonoids protect brain cells from damage due to oxidative stress.

### 2.6. Effects of Compounds ***1***, ***2***, ***5***, ***6***, ***8***, and ***10***–***12*** on Cell Viability through HO Signaling Pathway

Compounds **1**, **2**, **5**, **6**, **8**, and **10**–**12**, which exhibited cytoprotective effects, also increased HO-1 expression (Figure 3 and Figure 4). To investigate whether HO-1 expression regulates cell viability, we assessed the protective effects of compounds **1**, **2**, **5**, **6**, **8**, and **10**–**12** when tin protoporphyrin IX (SnPP) was used as a HO-1 activity inhibitor. Cells were treated with compounds **1**, **2**, **5**, **6**, **8**, and **10**–**12** (40 μM) in the presence or absence of SnPP (50 μM) and then exposed to glutamate (5 mM) for 12 h. When cells were pre-treated with SnPP, the protective effects of the compounds decreased (Figure 5); that is, cell viability was significantly lower in SnPP-pretreated cells than in the cells not treated with SnPP. These results indicate that compounds **1**, **2**, **5**, **6**, **8**, and **10**–**12** inhibited oxidative-stress-induced cell damage by increasing HO-1 protein expression.

## 3. Materials and Methods

### 3.1. Plant Materials

The flowers of *C. retusus* Lindl. And Paxton were gathered near Kyung Hee University, Yong-In, South Korea, in August 2014, and were identified by Prof. Dae-Keun Kim, College of Pharmacy, Woosuk University, Jeonju, South Korea. A voucher specimen (KHU-NPCl-201408) has been deposited at the Natural Products Chemistry Laboratory, Kyung Hee University.

### 3.2. General Experimental Procedures

The equipment and chemicals used to isolate and identify flavonoids from *C. retusus* flowers and evaluate their neuroprotective activity were obtained from the literature [32,33,34,35].

### 3.3. Isolation Procedure of Flavonoids *(**1***–***12**)* from C. retusus Flowers

Dried *C. retusus* flowers (315 g) were extracted in 80% aqueous MeOH (22.5 L × 4) at room temperature for 24 h, and then filtered and concentrated in vacuo. The concentrated MeOH extracts (145 g) were poured into H_2_O (2.0 L) and successively extracted with EtOAc (2.0 L × 3) and *n*-BuOH (1.8 L × 3). Each layer was concentrated under reduced pressure to obtain EtOAc (CFE, 27 g), *n*-BuOH (CFB, 24 g), and H_2_O (CFH, 94 g). Frs. CFE (27 g) was subjected to SiO_2_ c.c. (Φ 11 × 12 cm) and eluted with CHCl_3_‒MeOH (CM; 40:1 → 10:1 → 5:1 → 2:1 → 1:1, 600 mL of each), with monitoring by TLC, yielding 15 frs (CFE-1 to CFE-15).

CFE-5 (3.2 g, V*e*/V*t* 0.360–0.415) was subjected to ODS c.c. (Φ 5.5 × 7 cm, MeOH-H_2_O [MH] = 4:1, 1.7 L) to yield 12 Frs (CFE-5-1 to CFE-5-12). CFE-5-1 (1.0 g, V*e*/V*t* 0.000–0.110) was subjected to ODS c.c. (Φ 4.0 × 7 cm, MH = 1:1, 1.5 L) to yield 9 Frs (CFE-5-1-1 to CFE-5-1-9). CFE-5-1-3 (95.0 mg, V*e*/V*t* 0.150–0.260) was subjected to Sephadex LH-20 c.c. (Φ 1.5 × 60 cm, 80% MeOH, 560 mL) to yield 8 Frs (CFE-5-1-3-1 to CFE-5-1-3-8), along with purified compound **9** (CFE-5-1-3-4, 2.8 mg, V*e*/V*t* 0.550–0.560, TLC [SiO_2_] R_f_ 0.37, CM = 10:1, TLC [ODS] R_f_ 0.58, MH = 2:1).

CFE-7 (2.4 g, V*e*/V*t* 0.430–0.480) was subjected to Sephadex LH-20 c.c. (Φ 3 × 50 cm, 80% MeOH, 1.3 L) to yield 15 Frs (CFE-7-1 to CFE-7-15), along with purified compound **8** (CFE-7-10, 77.4 mg, V*e*/V*t* 0.488-0.542, TLC [SiO_2_] R_f_ 0.45, CHCl_3_-MeOH-H_2_O [CMH] = 10:3:1, TLC [ODS] R_f_ 0.60, MH = 3:2) and purified compound **1** (CFE-7-15, 14.6 mg, V*e*/V*t* 0.885-1.000, TLC [SiO_2_] R_f_ 0.47, CMH = 10:3:1, TLC [ODS] R_f_ 0.74, MH = 4:1). CFE-7-12 (68.5 mg, V*e*/V*t* 0.650–0.720) was subjected to ODS c.c. (Φ 2.0 × 7 cm, MH = 1:1, 620 mL) to yield 3 Frs (CFE-7-12-1 to CFE-7-12-3), along with purified compound **5** (CFE-7-12-2, 17.4 mg, V*e*/V*t* 0.194–0.677, TLC [SiO_2_] R_f_ 0.50, CMH = 10:3:1, TLC [ODS] R_f_ 0.50, MH = 4:1).

CFE-9 (2.2 g, V*e*/V*t* 0.580–0.610) was subjected to Sephadex LH-20 c.c. (Φ 3 × 50 cm, 80% MeOH, 2.2 L) to yield 14 Frs (CFE-9-1 to CFE-9-14). CFE-9-8 (36.5 mg, V*e*/V*t* 0.480-0.510) was subjected to ODS c.c. (Φ 2.0 × 5 cm, MH = 2:3, 200 mL) to yield 4 Frs (CFE-9-8-1 to CFE-9-8-4), along with purified compound **3** (CFE-9-8-2, 10.0 mg, V*e*/V*t* 0.125–0.425, TLC [SiO_2_] R_f_ 0.50, CM = 4:1, TLC [ODS] R_f_ 0.65, MH = 3:1).

CFE-12 (2.1 g, V*e*/V*t* 0.710-0.790) was subjected to Sephadex LH-20 c.c. (Φ 3.0 × 50 cm, 70% MeOH, 2.3 L) to yield 14 Frs (CFE-12-1 to CFE-12-14). CFE-12-5 (200.0 mg) was subjected to SiO_2_ c.c. (Φ 3.5 × 14 cm) and eluted with CMH = 10:3:1 (560 mL), with monitoring by TLC, yielding 6 Frs (CFE-12-5-1 to CFE-12-5-6), along with purified compound **10** (CFE-12-5-3, 119.4 mg, V*e*/V*t* 0.102–0.250, TLC [SiO_2_] R_f_ 0.50, CMH = 65:35:10, TLC [ODS] R_f_ 0.50, MH = 2:3). CFE-12-8 (330.0 mg, V*e*/V*t* 0.370–0.410) was subjected to ODS c.c. (Φ 3.0 × 5 cm, MH = 2:3, 1.2 L) to yield 8 Frs (CFE-12-8-1 to CFE-12-8-8), along with purified compound **12** (CFE-12-8-1, 115.5 mg, V*e*/V*t* 0.000–0.058, TLC [SiO_2_] R_f_ 0.50, CMH = 65:35:10, TLC [ODS] R_f_ 0.70, MH = 3:2). CFE-12-10 (240.0 mg, V*e*/V*t* 0.460-0.550) was subjected to ODS c.c. (Φ 3.0 × 5 cm, MH = 2:3, 840 mL) to yield 6 Frs (CFE-12-10-1 to CFE-12-10-6), along with purified compound **6** (CFE-12-10-4, 72.0 mg, V*e*/V*t* 0.286–0.414, TLC [SiO_2_] R_f_ 0.55, CMH = 65:35:10, TLC [ODS] R_f_ 0.45, MH = 3:2).

CFE-13 (3.2 g, V*e*/V*t* 0.710-0.790) was subjected to Sephadex LH-20 c.c. (Φ 3.0 × 50 cm, 70% MeOH, 2.3 L) to yield 16 Frs (CFE-13-1 to CFE-13-16), along with purified compound **2** (CFE-13-16, 29.0 mg, V*e*/V*t* 0.846–0.912, TLC [SiO_2_] R_f_ 0.50, CM = 5:1, TLC [ODS] R_f_ 0.40, MH = 3:1). CFE-13-6 (70.0 mg, V*e*/V*t* 0.270–0.320) was subjected to ODS c.c. (Φ 2.5 × 6 cm, MH = 2:3, 740 mL) to yield six Frs (CFE-13-6-1 to CFE-13-6-6), along with purified compound **7** (CFE-13-6-4, 12.0 mg, V*e*/V*t* 0.657–0.730, TLC [SiO_2_] R_f_ 0.50, CM = 2:1, TLC [ODS] R_f_ 0.55, MH = 3:2). CFE-13-7 (890.0 mg, V*e*/V*t* 0.330-0.480) was subjected to ODS c.c. (Φ 5.5 × 4 cm, MH = 2:3, 2.6 L) to yield 7 Frs (CFE-13-7-1 to CFE-13-7-7), along with purified compound **11** (CFE-13-7-1, 368.0 mg, V*e*/V*t* 0.000–0.102, TLC [SiO_2_] R_f_ 0.50, CM = 2:1, TLC [ODS] R_f_ 0.55, MH = 1:3) and compound **4** (CFE-13-7-6, 341.0 mg, V*e*/V*t* 0.512–0.923, TLC [SiO_2_] R_f_ 0.50, CM = 2:1, TLC [ODS] R_f_ 0.65, MH = 3:2) (Scheme 1).

*quercetin* (**1**): Yellowish powder (MeOH); m.p. 276–277 °C; ultraviolet (UV) (MeOH) λ_max_ (nm) 370, 305, 267, 255; infrared (IR) (KBr) ν_max_ 3350, 1680, 1615 cm^−1^; positive FAB/MS *m*/*z* 303 [M + H]^+^.

*kaempferol* (**2**): Yellowish powder (MeOH); m.p. 278–279 °C; UV (MeOH) λ_max_ (nm) 364, 320, 294, 265, 254; IR (KBr) ν_max_ 3345, 1658, 1605 cm^−1^; positive FAB/MS *m*/*z* 309 [M + Na]^+^.

*astragalin* (**3**): Yellowish powder (MeOH); m.p. 230–231 °C; ${\left[\alpha \right]}_{D}^{21}$+16.0 (*c* 0.1, MeOH); UV (MeOH) λ_max_ (nm) 348, 259; IR (KBr) ν_max_ 3350, 2930, 2365, 1655, 1610 cm^−1^; positive FAB/MS *m*/*z* 471 [M + Na]^+^.

*nicotiflorin* (**4**): Yellowish powder (MeOH); m.p. 268–269°C; ${\left[\alpha \right]}_{D}^{21}$−15.0 (*c* 1.0, MeOH); UV (MeOH) λ_max_ (nm) 365, 267, 254; IR (KBr) ν_max_ 3365, 2940, 2360, 1655, 1600, 1515 cm^−1^; positive FAB/MS *m*/*z* 639 [M + Na]^+^.

*luteolin* (**5**): Yellowish powder (MeOH); m.p. 329–330 °C; UV (MeOH) λ_max_ (nm) 349, 269, 254; IR (KBr) ν_max_ 3320, 2930, 1600, 1520 cm^−1^; positive FAB/MS *m*/*z* 271 [M + H]^+^.

*luteolin 4′-O-β-**d-glucopyranoside* (**6**): Yellowish powder (MeOH); m.p. 178–179 °C; UV (MeOH) λ_max_ (nm) 341, 272; IR (KBr) ν_max_ 3320, 2930, 1600, 1520, 1510, 1480 cm^−1^; positive FAB/MS *m*/*z* 449 [M + H]^+^.

*isorhoifolin* (**7**): Yellowish needles; m.p. 269–270 °C; ${\left[\alpha \right]}_{D}^{21}$−96.7 (*c* 1.0, MeOH); UV (MeOH) λ_max_ (nm) 331, 266; IR (KBr) ν_max_ 3365, 2360, 1635, 1600, 1515 cm^−1^; positive FAB/MS *m*/*z* 579 [M + H]^+^.

*taxifolin* (**8**): Yellowish powder (MeOH); m.p. 236–237 °C; ${\left[\alpha \right]}_{D}^{21}$+23.1 (*c* 0.1, MeOH); UV (MeOH) λ_max_ (nm) 330, 280; IR (KBr) ν_max_ 3415, 1625, 1515, 1472 cm^−1^; positive FAB/MS *m*/*z* 327 [M + Na]^+^.

*aromadendrin* (**9**): White powder; m.p. 216–217 °C; ${\left[\alpha \right]}_{D}^{21}$+58.5 (*c* 0.3, MeOH); UV (MeOH) λ_max_ (nm) 329, 292, 228; IR (KBr) ν_max_ 3420, 1655, 1518 cm^−1^; positive FAB/MS *m*/*z* 311 [M + Na]^+^.

*aromadendrin 7-O-β-**d-glucopyranoside* (**10**): Yellowish powder (MeOH); m.p. 172–173 °C; ${\left[\alpha \right]}_{D}^{21}$−18.7 (*c* 0.2, MeOH); UV (MeOH) λ_max_ (nm) 321, 285; IR (KBr) ν_max_ 3435, 1645, 1520, 1365 cm^−^^1^; positive FAB/MS *m*/*z* 473 [M + Na]^+^.

*taxifolin 7-O-β-**d-glucopyranoside* (**11**): Yellowish powder (MeOH); m.p. 169–170 °C; ${\left[\alpha \right]}_{D}^{21}$−48.2 (*c* 0.2, MeOH); UV (MeOH) λ_max_ (nm) 331, 283; IR (KBr) ν_max_ 3420, 1635, 1450, 1510, 1390 cm^−1^; positive FAB/MS *m*/*z* 467 [M + H]^+^.

*eriodictyol 7-O-β-**d-glucopyranoside* (**12**): Yellowish powder (MeOH); m.p. 173–174 °C; ${\left[\alpha \right]}_{D}^{21}$−35.5 (*c* 0.2, MeOH); UV (MeOH) λ_max_ (nm) 283, 233; IR (KBr) ν_max_ 3455, 1690, 1595, 1510 cm^−1^; positive FAB/MS *m*/*z* 451 [M + H]^+^.

^1^H-NMR (400 MHz, *δ*_H_) and ^13^C-NMR (100 MHz, *δ*_C_) spectroscopic data of flavonoids **1**–**12**, see Table 3 and Table 4.

### 3.4. Cell Culture and MTT Assay

Mouse hippocampal HT22 cells were donated by Wonkwang University, Iksan, Korea (Prof. Youn-Chul Kim). Cytoprotective activity assay was performed, as per the previously described method [35]. Cell viability was evaluated using the MTT assay reported in the literature [36].

### 3.5. Macrophage RAW 264.7 Culture, Viability Assay, and NO Measurement

Macrophage RAW 264.7 culture, viability assay, and NO measurement were carried out as per the previously described method [35].

### 3.6. Determination of Total Phenols and Flavonoids Contents in C. retusus Flower

Determination of the total phenolic and flavonoid contents of *C. retusus* flower was carried out as per the previously described method [37].

### 3.7. Western Blot Analysis

Pelleted HT22 cells were washed with PBS and lysed with an RIPA buffer from Sigma Chemical Co. The same amount of protein from each sample was mixed into a sample loading buffer, subjected to SDS-PAGE, and transferred to a membrane.

### 3.8. Statistical Analysis

Statistical analysis was performed with GraphPad Prism 5 software (ver. 3.03, San Diego, CA, USA). Data are presented as the mean ± standard deviation of 3 independent experiments. The mean differences were derived using one-way ANOVA and Tukey’s multiple comparison test, and statistical significance was defined as *p* < 0.05, *p* < 0.01, and *p* < 0.001.

## 4. Conclusions

In conclusion, four flavonols (**1**–**4**), three flavones (**5**–**7**), four flavanonols (**8**–**11**), and one flavanone (**12**) were isolated from *C. retusus* flowers. Flavonoids **4**–**6** and **8**–**11** were isolated from the flowers of *C. retusus* for the first time in this study. Flavonoids **1**, **2**, **5**, **6**, **8**, and **10**–**12** exhibited significant anti-inflammatory and neurocytoprotective activity, and effectively increased HO-1 protein expression. The flavonoids that displayed significant neuroprotective activity were found to recover oxidative stress-induced cell damage by increasing HO-1 protein expression. The relationships between the structural characteristics of these flavonoids and their anti-inflammatory and neuroprotective activity were revealed. Further studies are needed to investigate the potential therapeutic effects of flavonoids in innovative anti-inflammatory and neuroprotective strategies.

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
