# Peer review of "Flavonoids from Chionanthus retusus (Oleaceae) Flowers and Their Protective Effects against Glutamate-Induced Cell Toxicity in HT22 Cells"

_ijms, 2019, doi:10.3390/ijms20143517_

Reviewer 1 Report

Baek and co-workers reported the extraction, isolation and characterization of 12 flavonoids from Chionanthus retusus flowers.7 of them were found to significantly inhibiting NO production in RAW 264.7 stimulated by LPS and glutamate induced cell toxicity. Moreover they were found to inhibit oxidative-stress-induced cell damage by increasing HO-1 protein expression in mouse hippocampal HT22. Three of the active flavonoids were also isolated for the first time from C. retusus.

Major comments:

The presence of catechol structure in the B ring of flavonoids emerged as a key factor responsible for the effects of the studied flavonoids. Unfortunately cathecol is also classified as Pan-Assay Interference Compound (J.Med. Chem 2017, 60, 3879-3886). Some confirmatory assays have to be carried out in order to assess the activity of the reported compounds before publication.

Moreover the ability of the most active compounds to cross the blood brain barrier has to be investigated.

Minor comment:

The abbreviation for fractions have to be reported in line 63 not 71;

Author Response

Dear, Reviewer 1 of International Journal of Molecular Sciences

Thank you for your kind letter, with regard to our manuscript together with comments. We are thankful to you for the very valuable suggestions through the whole manuscript. Thank you again for your kind considerations.

We tried to revise the manuscript as much as possible in line with suggestion made by the editor and reviewers. The corrected was expressed in blue. I am herewith enclosing improved manuscript.

Our incorporation of editor and reviewer 1’s suggestions is as follow.

- Reviewer 1

Baek and co-workers reported the extraction, isolation and characterization of 12 flavonoids from Chionanthus retusus flowers.7 of them were found to significantly inhibiting NO production in RAW 264.7 stimulated by LPS and glutamate induced cell toxicity. Moreover they were found to inhibit oxidative-stress-induced cell damage by increasing HO-1 protein expression in mouse hippocampal HT22. Three of the active flavonoids were also isolated for the first time from C. retusus.

Major comments:

1) The presence of catechol structure in the B ring of flavonoids emerged as a key factor responsible for the effects of the studied flavonoids. Unfortunately cathecol is also classified as Pan-Assay Interference Compound (J.Med. Chem 2017, 60, 3879-3886). Some confirmatory assays have to be carried out in order to assess the activity of the reported compounds before publication.

Moreover the ability of the most active compounds to cross the blood brain barrier has to be investigated.

  Thank you for your meaningful comment. According to referenece 1-3, they are suggesting that flavonoids with catechol structures pass through the blood brain barrier (BBB) with a neuroprotective effect We did this on the basis of these studies. When we do this kind of experiment, we will assure to refer to the previous study more carefully and to examine whether it crosses BBB from now on as you indicated. Please we hope that this has been appropriate and acceptable answer to your comment.

References

1. Faria, A., Meireles, M., Fernandes, I., Santos-Buelga, C., Gonzalez-Manzano, S., Dueñas, M., De Freitas, V., Mateus, N., Calhau, C. Flavonoid metabolites transport across a human BBB model. Food Chem 2014, 149, 190-196.

2. Ren, S.C., Suo, Q.F., Du, W.T., Pan, H., Yang, M.M., Wang, R.H., Liu, J. Quercetin permeability across blood-brain barrier and its effect on the viability of U251 cells. J Sichuan Univ 2010, 41, 751-754.

3. Dok-Go, H., Lee, K.H., Kim, H.J., Lee, E.H., Lee, J., Song, Y.S., Lee, Y.H., Jin, C., Lee, Y.S., Cho, J. Neuroprotective effects of antioxidative flavonoids, quercetin,(+)-dihydroquercetin and quercetin 3-methyl ether, isolated from Opuntia ficus-indica var. saboten. Brain Res 2003, 965, 130-136.

Minor comment:

The abbreviation for fractions have to be reported in line 63 not 71;

Thank you for your comment. As indicated, the abbreviation was revised as indicated.

I hope the improved version will be acceptable for publication in International Journal of Molecular Sciences.

Yours sincerely

Prof. Nam-In Baek

Reviewer 2 Report

In this manuscript, Lee et al., showed the protective effects of Chionanthus retusus against glutamate-induced toxicity using neuronal cell line. However, this study is limited to the cell line HT22 with SV40 T-antigen. In many cases, this manuscript should be significantly improved.

Major comments

1. I strongly recommend that the authors should assess the neuroprotective function of Chionanthus retusus using primary culture hippocampal neurons or in vivo experiments, since they claim “neuroprotective” effects. 

2. The authors should reveal the downstream molecular mechanism of reduced neurotoxicity induced by Chionanthus retusus for this manuscript to be appropriate for publication in this journal. Using siRNA-mediated knockdown method would be helpful, since inhibitors have nonspecific effects.

3. The authors showed Chionanthus retusus decreased HO-1 expression. If HO-1 is involved in neuroprotective function of Chionanthus retusus, loss of function experiments (e.g. HO-1 siRNA) in primary neurons are required.

Minor comments

1. Cellular viabilities were examined only using MTT assay. How was the effect of Chionanthus retusus on cellular proliferation? This point should be assessed.

2. Quantitative analysis is required for Fig. 3.

3. line 116, 118: Fig. 3 needed to be changed to Fig. 2?

4. What statistical tests used for the quantitative analysis in Fig. 2 and 4. Since the same control group seems to be used for various treatment, Anova should be used.

Author Response

Dear, Reviewer 2 of International Journal of Molecular Sciences

Thank you for your kind letter, with regard to our manuscript together with comments. We are thankful to you for the very valuable suggestions through the whole manuscript. Thank you again for your kind considerations.

We tried to revise the manuscript as much as possible in line with suggestion made by the editor and reviewers. The corrected was expressed in blue. I am herewith enclosing improved manuscript.

Our incorporation of editor and reviewer 2’s suggestions is as follow.

- Reviewer 2

In this manuscript, Lee et al., showed the protective effects of Chionanthus retusus against glutamate-induced toxicity using neuronal cell line. However, this study is limited to the cell line HT22 with SV40 T-antigen. In many cases, this manuscript should be significantly improved.

Major comments

1) I strongly recommend that the authors should assess the neuroprotective function of Chionanthus retusus using primary culture hippocampal neurons or in vivo experiments, since they claim “neuroprotective” effects.

Thank you for your comment. The purpose of this study were isolation and identification of flavonoids from the flowers of Chionanthus retusus along with suggestion futher use of these compounds and its extract. And one of them was to present the potential for use in degenerative brain diseases. Through follow-up research, we will study detailed mechanisms using primary culture hippocampal neurons or in vivo experiments. Please we hope that this has been appropriate and acceptable answer to your comment.

2) The authors should reveal the downstream molecular mechanism of reduced neurotoxicity induced by Chionanthus retusus for this manuscript to be appropriate for publication in this journal. Using siRNA-mediated knockdown method would be helpful, since inhibitors have nonspecific effects.

Thank you for your comment. I really agree to your indication. But our research team are difficult to conduct experiments experiment with in vivo or primary cells. In order to carry out such experiments, we need to conduct joint-research with a professional research team, but we have not found it at this time. Through follow-up research, we will study detailed mechanisms with specialized research teams. Please we hope that this has been appropriate and acceptable answer to your comment.

3) The authors showed Chionanthus retusus decreased HO-1 expression. If HO-1 is involved in neuroprotective function of Chionanthus retusus, loss of function experiments (e.g. HO-1 siRNA) in primary neurons are required.

Thank you for your comment. I really agree to your indication. The amount of almost isolated flavonoids in this study is lack to do additional experiments with in vivo or using primary cells. In current, additional separation of flavonoids from Chionanthus retusus is being to be carried out for further study. Therefore follow-up research, we will study experiments (e.g. HO-1 siRNA) in primary neurons after sufficient quantity is obtained. Please we hope that this has been appropriate and acceptable answer to your comment.

Minor comments

1) Cellular viabilities were examined only using MTT assay. How was the effect of Chionanthus retusus on cellular proliferation? This point should be assessed.

Thank you for your meaningful comment. It is well-known that measurement of cell viability and proliferation forms the basis for numerous in vitro assays of a cell population’s response to external factors [1-5]. The reduction of tetrazolium salts is now widely accepted as a reliable way to examine cell proliferation. The yellow tetrazolium MTT (3-(4, 5-dimethylthiazolyl-2)-2, 5-diphenyltetrazolium bromide) is reduced by metabolically active cells, in part by the action of dehydrogenase enzymes, to generate reducing equivalents such as NADH and NADPH. The resulting intracellular purple formazan can be solubilized and quantified by spectrophotometric means [1-5]. Therefore, many of the MTT Cell Proliferation Assay Kit or MTT assay experiment measures the cell proliferation rate and conversely, when metabolic events lead to apoptosis or necrosis, the reduction in cell viability [1-5]. For these reason, we have used MTT assay for the determination of Cellular viabilities.

In addition, according to your comment, we have added the experiments on the Supple Figure 1 about the effect of cellular proliferation or cytotoxic effects by compounds 1-12 using MTT assay. To investigate the cytotoxic potential of compounds 1-12, we first examined the viability of mouse hippocampal HT22 cells treated with compounds 1-12 using the MTT assay No cytotoxic effects or cellular proliferation by compounds 1-12 were observed at concentrations<40 µM (Supple Fig. 1). We have added the above point in the results section. Please we hope that this has been an appropriate answer and experiment to your comment.

References

1. van de Loosdrecht, A.A., et al. J Immunol Methods 1994; 174, 311-320.

2. Ferrari, M., et al. J Immunol Methods 1990; 131: 165-172.

3. Gerlier, D., and N. Thomasset. J Immunol Methods 1986; 94: 57-63.

4. Alley, M.C., et al. Cancer Res 1988; 48: 589-601.

5. Mosmann, T. J Immunol Methods1983; 65: 55-63.

2) Quantitative analysis is required for Fig. 3.

Thank you for your comment. According to your comment, we have added the quantitative analysis of WB in the Fig. 3.

3) line 116, 118: Fig. 3 needed to be changed to Fig. 2?

Thank you for your comment. As indicated, Fig. 3. in line 116 and 118 were changed to Fig. 2.

4) What statistical tests used for the quantitative analysis in Fig. 2 and 4. Since the same control group seems to be used for various treatment, Anova should be used.

Thank you for your important comment. Actually, we already have used the mean differences were derived using one-way ANOVA and Tukey’s multiple comparison test. However, we have not described about the detail methods of Statistical analysis in the manuscript. It is our mistake, so we are sorry about this point. After your comment, we have added the detail methods of Statistical analysis in the manuscript, and we have checked the Statistical analysis one more time. In addition, we have revised the figure 2 and 4 about quantitative analysis especially p-value. It is easy to understand more the comparison between each group. Please we hope that this has been an appropriate answer and experiment to your comment.

I hope the improved version will be acceptable for publication in International Journal of Molecular Sciences.

Yours sincerely

Prof. Nam-In Baek

Round  2

Reviewer 1 Report

The manuscript of Lee et al. describes the neuroprotective effects of some flavonoids from Chionantus retusus. The reported conclusions are sufficiently supported by the experimental results. 

I suggest the publication after minor revisions.

Some comments on the toxicity of compounds 1-12 on RAW264.7 cells have to be added as well as  the authors have done for the experiment carried out on HT22 cells. 

The English language needs to be checked.

Author Response

 Dear, Editor in Chief, International Journal of Molecular Sciences

Thank you for your kind letter, with regard to our manuscript together with comments. We are thankful to you for the very valuable suggestions through the whole manuscript. Thank you again for your kind considerations.

We tried to revise the manuscript as much as possible in line with suggestion made by the reviewer 1. The corrected was expressed in blue. I am herewith enclosing improved manuscript.

Our incorporation of reviewer 1’s suggestions is as follow.

- Reviewer 1

The manuscript of Lee et al. describes the neuroprotective effects of some flavonoids from Chionantus retusus. The reported conclusions are sufficiently supported by the experimental results.

I suggest the publication after minor revisions.

Some comments on the toxicity of compounds 1-12 on RAW264.7 cells have to be added as well as the authors have done for the experiment carried out on HT22 cells.

Thank you for your meaningful comment. According to your comment, we checked the effect of by compounds 1-12 for proliferation or cytotoxic using MTT assay. We first examined the viability of RAW264.7 cells treated with compounds 1-12. No cytotoxic effects or cellular proliferation by compounds 1-12 were observed at concentrations<40 µM (Table 2), which was added in the results section

The English language needs to be checked.

Actually, this manuscript was checked by a company including native speakers prior to submission (The proof document is added). But we did our best to make the better writing.    

I hope the improved version will be acceptable for publication in International Journal of Molecular Sciences.

Yours sincerely

Prof. Nam-In Baek

Reviewer 2 Report

The authors have addressed most of the issues that I raised in my previous review but a few still remain. 

Major comment

(1) If the authors do not intend to add the data using neuron, they should not use the term "neuroprotective" since HT-22 cells are not neurons.

Author Response

Dear, Reviewer 2

Thank you for your kind letter, with regard to our manuscript together with comments. We are thankful to you for the very valuable suggestions through the whole manuscript. Thank you again for your kind considerations.

We tried to revise the manuscript as much as possible in line with suggestion made by reviewer 2. The corrected was expressed in blue. I am herewith enclosing improved manuscript.

Our incorporation of editor and reviewer 2’s suggestions is as follow. 

- Reviewer 2

The authors have addressed most of the issues that I raised in my previous review but a few still remain.

Major comment

(1) If the authors do not intend to add the data using neuron, they should not use the term "neuroprotective" since HT-22 cells are not neurons.

Thank you for your comment. As indicated, the title of this manuscript was revised as indicated.

Flavonoids from Chionanthus retusus (Oleaceae) flowers and their protective effects against glutamate-induced cell toxicity through heme oxygenase-1 expression in mouse hippocampal HT22 cells

I hope the improved version will be acceptable for publication in International Journal of Molecular Sciences.

Yours sincerely

Prof. Nam-In Baek